# Commuting to University: Self-Reported and Device-Measured Physical Activity and Sedentary Behaviour



Ximena Palma-Leal [1,2], Palma Chillón [1,*], Víctor Segura-Jiménez [3,4,5], Alejandro Pérez-Bey [5,6], Alejandro Sánchez-Delgado [5,6] and Daniel Camiletti-Moirón [5,6]

1 PROFITH Research Group, Department of Physical Education and Sports, Faculty of Sport Sciences, Sport and Health University Research Institute (iMUDS), University of Granada, 18011 Granada, Spain
2 IRyS Group, School of Physical Education, Pontificia Universidad Católica de Valparaíso, Viña del Mar 2340000, Chile
3 Hospital Universitario Virgen de las Nieves, 18011 Granada, Spain
4 Instituto de Investigación Biosanitaria ibs. GRANADA, 18011 Granada, Spain
5 GALENO Research Group, Department of Physical Education, Faculty of Education Sciences, University of Cádiz, Avenida República Saharaui s/n, Puerto Real, 11519 Cádiz, Spain
6 Instituto de Investigación e Innovación Biomédica de Cádiz (INiBICA), 11009 Cádiz, Spain
* Correspondence: pchillon@ugr.es; Tel.: +34-958244353

**Abstract:** Background: Active commuting could provide an opportunity to counteract unhealthy behaviours, such as insufficient levels of Physical Activity (PA) and sedentary behaviour, which are major health problems in the university population. The aims of this study were to describe and compare self-reported and device-measured in commuting behaviours, PA, and sedentary behaviour in both trips (to and from university) by mode of commuting per weekday, and to identify associations between self-reported and device-measured of commuting behaviours, PA, and sedentary behaviour. Methods: After inclusion criteria, a total of 63 students (65.1% women) from a public university in Cádiz, Spain, participated in this study. Self-reported and device-measured information was used. Results: Commuting time, distance, and speed were lower in active commuters than public and private commuters in both trips (to and from university) (all, $p < 0.001$). Commuting energy expenditure per min was higher in active commuters than public and private commuters (all, $p < 0.001$). Active commuters presented significant differences ($p < 0.05$) with public and private commuters in all PA levels and sedentary behaviour in both trips (to and from university). Conclusions: Active commuting involved the highest levels of energy expenditure per min and could contribute 44% of the weekly PA recommendation for health benefits in university students.

**Keywords:** physical activity; active travel; sedentary behaviour; college students





## 1. Introduction

Despite there being irrefutable evidence that physical activity (PA) is beneficial to health [1], the prevalence of being physically active remains low, with an especially significant reduction during the university period [2]. University students are in positions of vulnerability and influence in this new stage of life [3], which is characterized by long days of study, little time dedicated to PA, and high sitting time [4]. Regarding this, a study in university students from 24 countries showed that students with higher sedentary behaviour (≥8 h sitting time) had lower PA levels [5]. In Spain, it is estimated that a small number of university students meet the recommendation of moderate-to-vigorous PA (MVPA) [6]. Therefore, useful strategies for promoting PA and reducing sedentary behaviour among university students are recommended.

Active commuting (understood as a frequent travel from one place to another by the physical effort of a person, being the most common modes walking and cycling) provides

an opportunity to increase PA levels in university students [7] and reduce sedentary behaviour [8,9]. Furthermore, active commuting has been considered and differentiated from the motorised modes, such as public commuting (e.g., public bus, metro, train) or private commuting (e.g., car, motorcycle), as a more sustainable transportation, having a direct impact on reducing parking demand on campuses and helping to make the environment cleaner due to lower emissions [10]. However, it is important to highlight that comparing motorised modes, it has been found that public commuting could help maintain PA levels (due to higher levels of daily steps) [11] and could reduce congestion and pollution [12] compared to private commuting. Therefore, if the use of motorised modes is the only option (e.g., due to time, distance, weather, etc.), the recommendation could be to use public over private commuting.

An important aspect of the modes of commuting and their behaviours (such as time or distance) is that they have been mainly assessed by self-reported measures. Although self-reported measures are useful for identifying commuting modes, they do not objectively assess the contribution of modes of commuting to daily PA levels [13] and sedentary behaviour. A global study including data from 49 countries in 5 continents indicated that to better assess to what extent the active commuting contributes to PA, commuting behaviours should be measured [14]. Device measures, such as accelerometers, may provide accurate information regarding frequency, intensity, and the amount of time spent in PA and sedentary behaviour [15]. For instance, 12 studies evaluated PA related to commuting to school using accelerometer-based devices and showed that active commuting contributed to 23% and 36% of MVPA per day in children and adolescents, respectively [16]. Nevertheless, it has been shown that accelerometers have a filtering function that could eliminate activity counts for signal frequencies above 2.5 Hz, which are the frequencies generated by the most vigorous PA [17], and they are unable to capture some types of movements, such as cycling [18]. In this line, a longitudinal study in adolescents that used self-reports and accelerometer-based devices showed that there are different effects in the same analyses (significant vs. non-significant) depending on the model used (self-reports vs accelerometer-based devices) [19]. The evidence suggests that self-reported and device- measured should be seen as complementary instruments [20] but that the information should be studied with caution and not used interchangeably.

The evidence of the commuting-related PA and sedentary behaviour in university students is still scarce, and it is important to take into account the implications of university life (off-campus days, varying class times per day, evening study times, etc.). Therefore, the aims of this study were (a) to describe and compare self-reported and device-measured in commuting behaviours, PA, and sedentary behaviour in both trips (to and from university) by mode of commuting per weekday, and (b) and to identify associations between self-reported and device-measured of commuting behaviours, PA, and sedentary behaviour.

## 2. Materials and Methods

### 2.1. Study Design and Participants

This observational cross-sectional study used information from the UCActive study (patterns of commuting and PA in university students and staff of the University of Cádiz) and was conducted from March to May 2018. The present study included a total sample size of 99 university students. Participants who did not provide complete data (n = 30), and those who did not meet the accelerometery inclusion criteria (n = 6) were excluded. A total of 36 participants were excluded, and the final sample included 63 university students (65.1% women), with an average age of 20.6 ± 3.8 years.

The participants belonged to different degree programmes from two campuses of the University of Cádiz. The samples were recruited by convenience. All interested university students were invited to voluntarily participate in this study and received detailed information (e.g., methods, objectives), and those who agreed to participate signed an informed consent form.

## 2.2. Procedures

Firstly, an online invitation was made on the university platform. Secondly, the faculties were visited in person to give more visibility to the details of the project. Finally, university students interested in participating were invited to the Physiology Laboratory of the Faculty of Education Sciences of the University of Cádiz for the evaluations.

The university students were evaluated twice. At the first visit, weight and height were measured. Participants were asked to wear an accelerometer for 8 consecutive days, starting the same day they received the device. Participants attended a second visit to complete the self-reported online-based questionnaire and return the accelerometers to the researchers.

## 2.3. Self-Reported Measures

The questionnaire used included questions about demographic variables and commuting behaviours and was presented as reliable for university students [21]. Participants self-reported sociodemographic characteristics, such as age, sex, current postal address as a student, type of residence, and socioeconomic characteristics. The type of residence was assessed using the question "With whom do you live?", and answer options were divided into two categories: family residence (e.g., parents' home or own house) and university residence (e.g., shared flat with other students or hall of residence). Finally, for socioeconomic levels, the Family Affluence Scale was used [22], classified into three categories: low (0 to 3 points), medium (4 to 5 points), and high (6 to 7 points).

### 2.3.1. Mode of Commuting

This includes two separate questions: (1) how do you usually travel to university? and, (2) how do you usually travel from university? The answer options were walking, cycling, car, motorcycle, public bus, metro, train, and no travel. Participants were classified into three categories as: "active" (walking), "public" (public bus, metro, and train), and "private" (car and motorcycle) commuting [23]. The basis of this classification is the different PA levels that could be represented by each mode of commuting (from highest to lowest PA levels, respectively) [24].

### 2.3.2. Commuting Behaviours Variables

All the details of the information related to commuting behaviour variables have been described elsewhere [7]. Product of a line of research on university students commuting with databases from Latin American and European countries, of which the present study is part. Briefly, based on self-reported data, we considered calculating variables to provide a better understanding of commuting behaviour. The commuting variables were: *commuting time* (self-reported daily min); *commuting distance* (geocoded by the research team); *commuting speed* (based on commuting distance/commuting time); and two *commuting energy expenditures: per min and total* (shown in metabolic equivalents (METs) and based in the code of the Compendium of Physical Activities for adults [25], see calculation examples: Supplementary Material Table S1).

## 2.4. Device Measures

The ActiGraph accelerometer GT3X+ (ActiGraph, Pensacola, FL, USA) was used to measure PA levels and sedentary behaviour. Data were collected with the low-frequency extension filter disabled at a sampling frequency of 60 Hz and subsequently collapsed to 60 s epochs. Data from ActiGraph accelerometers were downloaded and processed using the ActiLife software v6.13.3. Non-wear periods were identified by applying the algorithm developed by Choi et al. [26] (e.g., bouts of 90 continuous min (30 min minimum up/down stream time for consecutive zero counts, and a 2 min skip tolerance) with no data (counts) were considered non-wear time and excluded from the analyses). From the full-day data, only recordings during the commuting to and from university were studied, based on the completed diary. Participants were instructed to wear the accelerometer around their

hips attached by an elastic belt over the whole day (24 h), for 8 days and were advised to keep on with their usual lifestyle. To protect the accelerometers, participants were asked to take them off while bathing or swimming. After these instructions, the students were asked to commit to comply and to take care of the device. Accelerometer-wearing time was obtained by subtracting sleeping time (which was obtained from a diary in which participants indicated sleep and wake-up times) and non-wear periods from each day. PA levels and sedentary behaviour during the time of commuting to and from university were set as the time (min/journey) engaged in sedentary, light, moderate, vigorous, and MVPA based upon standardised cut-offs of 0–200, 200–2689, 2690–6166, and $\geq$6167 counts per min, respectively [27,28] Vigorous PA was excluded from the tables and the figure since its median value was generally zero in the three modes of commuting.

### 2.5. Body Composition

Weight was measured with an electronic scale (SECA 861 Hamburg, Germany; range, 0.05–130 kg; precision, 0.05 kg) and height (measured in the Frankfort plane) with a telescopic stature-measuring instrument (Type SECA 225; range, 60–200 cm; precision, 1 mm). Body mass index (BMI) was calculated as weight/height squared (kg/m$^2$). Measurements were performed twice, and the mean value of the two measurements was used in the analyses.

### 2.6. Statistical Analysis

Descriptive characteristics were analysed using descriptive statistics and were reported as the mean $\pm$ standard deviation (SD) for continuous variables and as frequencies and percentages (%) for categorical variables. The differences in the descriptive characteristics between each mode of commuting were analysed using chi-square test and standard analysis of variance (ANOVA) for categorical and continuous variables, respectively. Post hoc analysis with Bonferroni's correction was used due to the large number of comparisons between the groups and to ascertain differences between them (active vs. public commuting, active vs. private commuting, and public vs. private commuting). Self-reported and device-measured of commuting behaviours, PA, and sedentary behaviour in both trips (to and from university) by mode of commuting, were presented as medians and interquartile ranges (IQRs (Q3–Q1)). The differences in self-reported and device-measured of commuting behaviours, PA, and sedentary behaviour were analysed using the Kruskal–Wallis test. Additionally, to ascertain differences between groups the Mann–Whitney test was used. Finally, associations between self-reported and device-measured of commuting behaviours, PA, and sedentary behaviour were studied using linear regression. Self-reported measures to and from university were included in the model as dependent variables, and the device-measured data were included as independent variables individually in separate models for each mode of commuting. To ascertain whether "to" and "from" university differed, a paired t-test was carried out, which turned out to be significant, therefore the results were shown separately. The level of significance in all analyses was set to $p < 0.05$. The statistical analyses were conducted using the IBM SPSS Statistics (v.25.0 for WINDOWS, Chicago, IL, USA).

### 3. Results

Table 1 shows the descriptive characteristics of participants by the mode of commuting to university. The main mode of commuting used was private. Most of the sample were women (65.1%) and had medium socioeconomic status (58.7%). Students living in university residence were mostly active commuters ($p = 0.002$). There were no significant differences in any body composition variables (all $p > 0.05$).

**Table 1.** Descriptive characteristics of participants by the mode of commuting to university.

| | Mode of Commuting to University | | | | |
|---|---|---|---|---|---|
| | **All** n = 63 (100) | **Active** n = 18 (28.6) | **Public** n = 13 (20.6) | **Private** n = 32 (50.8) | ***p*-Value** |
| **Demographic characteristics** | | | | | |
| Age | 20.6 ± 3.8 | 19.9 ± 1.2 | 22.1 ± 8.1 | 20.4 ± 1.6 | 0.283 |
| Sex | | | | | |
| Men | 22 (34.9) | 7 (38.9) | 4 (26.7) | 11 (36.7) | 0.735 |
| Women | 41 (65.1) | 11 (61.1) | 11 (73.3) | 19 (63.3) | |
| Socioeconomic levels | | | | | |
| High | 18 (28.6) | 4 (22.2) | 2 (13.3) | 12 (40.0) | |
| Medium | 37 (58.7) | 12 (66.7) | 9 (60.0) | 16 (53.3) | 0.172 |
| Low | 8 (12.7) | 2 (11.1) | 4 (26.7) | 2 (6.7) | |
| Type of residence | | | | | |
| Family residence | 40 (63.5) | 6 (33.3) [a] | 9 (60.0) | 25 (83.3) [a] | 0.002 |
| University residence | 23 (36.5) | 12 (66.7) [a] | 6 (40.0) | 5 (16.7) [a] | |
| **Body composition** | | | | | |
| Height (cm) | 165.4 ± 8.5 | 165.9 ± 7.4 | 162.4 ± 9.0 | 166.3 ± 8.8 | 0.683 |
| Weight (kg) | 61.8 ± 11.6 | 62.7 ± 9.2 | 62.8 ± 18.3 | 60.8 ± 9.6 | 0.686 |
| BMI (kg × m$^2$) | 22.4 ± 2.8 | 22.7 ± 2.3 | 23.4 ± 4.6 | 21.8 ± 2.1 | 0.234 |

Notes: The data are reported as mean ± standard deviation for continuous variables and as frequency (%) for categorical variables; cm = centimetres; kg = kilograms; kg × m$^2$ = kg/m$^2$; BMI = body mass index; and common superscripts indicate significant differences ($p < 0.05$) between the groups with the same letter after Mann–Whitney test.

Table 2 presents the medians and interquartile ranges of the self-reported measures of commuting behaviours to and from the university. In both trips (to and from university), all the commuting behaviour variables (commuting time, distance, and speed) were lower for active commuters than for both motorised modes, except for commuting energy expenditure per min (all, $p < 0.001$). Public commuters present statistical differences from active and private commuters in the total commuting energy expenditure ($p < 0.05$).

**Table 2.** Self-reported measures of commuting behaviours to and from the university of the participants.

| | All (n = 63) | | | |
|---|---|---|---|---|
| | **To University** | ***p*-Value** | **From University** | ***p*-Value** |
| **Commuting Behaviours** | **Median (IQRs)** | | **Median (IQRs)** | |
| Commuting time (min) | | | | |
| Active | 9.5 (3.0) [a,b] | | 10.0 (3.0) [a,b] | |
| Public | 25.0 (30.0) [a] | <0.001 | 25.0 (25.0) [a] | <0.001 |
| Private | 25.0 (18.7) [b] | | 25.0 (20.0) [b] | |
| Commuting distance (km) | | | | |
| Active | 0.7 (0.3) [a,b] | | 0.7 (0.3) [a,b] | |
| Public | 11.0 (19.3) [a] | <0.001 | 10.1 (10.8) [a] | <0.001 |
| Private | 18.2 (22.4) [b] | | 19.3 (21.7) [b] | |
| Commuting speed (km/hr) | | | | |
| Active | 4.8 (1.2) [a,b] | | 4.8 (1.2) [a,b] | |
| Public | 27.3 (13.6) [a] | <0.001 | 26.2 (15.4) [a] | <0.001 |
| Private | 45.5 (31.3) [b] | | 46.0 (32.4) [b] | |
| EE per min (METs) | | | | |
| Active | 3.7 (0.9) [a,b] | | 3.7 (0.3) [a,b] | |
| Public | 2.1 (0.5) [a,c] | <0.001 | 2.1 (0.5) [a,c] | <0.001 |
| Private | 1.3 (0.0) [b,c] | | 1.3 (0.0) [b,c] | |

**Table 2.** *Cont.*

| Commuting Behaviours | To University | p-Value | From University | p-Value |
|---|---|---|---|---|
| | **All (n = 63)** | | | |
| | Median (IQRs) | | Median (IQRs) | |
| Total EE (METs) | | | | |
| Active | 30.1 (11.5) [a] | | 30.1 (10.7) [a] | |
| Public | 51.2 (39.0) [a,b] | 0.001 | 51.2 (32.5) [a,b] | 0.006 |
| Private | 32.5 (24.3) [b] | | 32.5 (26.0) [b] | |

Notes: IQRs = interquartile ranges [Q3–Q1]; min = minutes; km = kilometres; hr = hours; EE = energy expenditure; METs = metabolic equivalents; and common superscripts indicate significant differences ($p < 0.05$) between the groups with the same letter after Mann–Whitney test.

Table 3 shows the medians and interquartile ranges of device-measured time in PA levels and sedentary behaviour per weekday in both trips (to and from university) by mode of commuting. Overall, to and from university, there were significant differences in MVPA level and light PA level between active commuters and both motorised modes ($p < 0.05$). Sedentary behaviour presented significant differences ($p < 0.05$) to and from university on weekdays (except on Thursday from university) between active commuters vs public and/or private commuters, and the medians of active commuters were always close to zero (vs public and/or private commuting with medians between 10 to 34 min).

**Table 3.** Device-measured time in PA levels and sedentary behaviour in both trips (to and from university) by mode of commuting per weekday of the participants.

| Commuting-Related PA and SB (min) | Active n = 18 (28.6) | Public n = 13 (20.6) | Private n = 32 (50.8) | p-Value | Active n = 18 (28.6) | Public n = 14 (22.2) | Private n = 31 (49.2) | p-Value |
|---|---|---|---|---|---|---|---|---|
| | **Mode of Commuting to University** | | | | **Mode of Commuting from University** | | | |
| | Median (IQRs) | Median (IQRs) | Median (IQRs) | | Median (IQRs) | Median (IQRs) | Median (IQRs) | |
| **Monday** | | | | | | | | |
| MVPA | 8.5 (4.2) [a,b] | 2.0 (6.0) [a] | 1.0 (3.0) [b] | <0.001 | 6.0 (7.0) [a,b] | 2.5 (5.0) [a] | 0.0 (3.0) [b] | <0.001 |
| Moderate | 8.0 (5.2) [a,b] | 2.0 (6.0) [a] | 1.0 (2.7) [b] | 0.001 | 5.5 (6.7) [a,b] | 2.5 (5.0) [a] | 0.0 (3.0) [b] | <0.001 |
| Light | 2.0 (3.5) [a,b] | 8.0 (11.0) [a] | 8.5 (9.2) [b] | <0.001 | 4.0 (8.0) [b] | 10.5 (3.5) | 12.0 (9.0) [b] | 0.020 |
| Sedentary | 0.0 (2.0) [a,b] | 19.0 (14.0) [a] | 18.0 (15.2) [b] | <0.001 | 2.0 (5.2) [a,b] | 17.0 (11.5) [a] | 12.0 (18.0) [b] | <0.001 |
| **Tuesday** | | | | | | | | |
| MVPA | 7.0 (4.0) [a,b] | 1.0 (5.7) [a] | 1.0 (3.0) [b] | <0.001 | 7.0 (4.5) [b] | 2.0 (6.5) | 1.0 (4.2) [b] | 0.005 |
| Moderate | 5.0 (4.0) [a,b] | 1.0 (5.7) [a] | 1.0 (3.0) [b] | 0.001 | 7.0 (4.0) [b] | 2.0 (6.5) | 1.0 (4.2) [b] | 0.005 |
| Light | 3.0 (5.0) [a,b] | 7.5 (9.5) [a] | 8.0 (8.0) [b] | 0.001 | 6.0 (7.5) [b] | 8.0 (8.0) | 11.0 (13.2) [b] | 0.009 |
| Sedentary | 0.0 (0.0) [a,b] | 23.5 (12.5) [a] | 15.0 (16.0) [b] | <0.001 | 1.0 (6.0) [a,b] | 15.0 (19.0) [a] | 18.0 (12.2) [b] | <0.001 |
| **Wednesday** | | | | | | | | |
| MVPA | 7.0 (5.7) [b] | 2.1 (22.0) | 1.0 (2.0) [b] | 0.003 | 7.0 (11.0) [b] | 1.6 (11.5) | 0.5 (4.7) [b] | 0.050 |
| Moderate | 4.5 (7.5) [b] | 2.0 (19.0) | 1.0 (2.0) [b] | 0.010 | 6.0 (12.0) [b] | 1.5 (5.5) | 0.5 (4.7) [b] | 0.058 |
| Light | 2.0 (4.0) [a,b] | 8.0 (5.0) [a] | 9.0 (10.0) [b] | <0.001 | 7.0 (8.0) | 10.0 (6.7) | 9.5 (9.5) | 0.281 |
| Sedentary | 0.0 (2.5) [a,b] | 24.0 (15.0) [a] | 17.0 (12.5) [b] | <0.001 | 2.0 (5.5) [a,b] | 14.0 (8.2) [a] | 15.5 (17.5) [b] | 0.006 |
| **Thursday** | | | | | | | | |
| MVPA | 7.0 (15.5) [a,b] | 2.0 (3.5) [a] | 1.0 (2.0) [b] | 0.001 | 9.0 (20.5) | 0.0 (2.0) | 2.0 (4.0) | 0.287 |
| Moderate | 7.0 (15.5) [a,b] | 2.0 (3.5) [a] | 1.0 (2.0) [b] | 0.001 | 9.0 (20.5) | 0.0 (2.0) | 2.0 (4.0) | 0.287 |
| Light | 2.0 (5.5) [b] | 10.0 (10.5) | 10.0 (9.0) [b] | 0.042 | 0.0 (3.5) [b] | 5.0 (2.0) [c] | 11.0 (10.0) [b,c] | 0.001 |
| Sedentary | 0.0 (0.0) [a,b] | 19.0 (12.0) [a] | 13.0 (16.0) [b] | 0.002 | 1.0 (7.5) [a,b] | 15.0 (16.0) [a] | 20.0 (23.0) [b] | 0.013 |
| **Friday** | | | | | | | | |
| MVPA | 13.0 (6.0) [a,b] | 1.5 (0.0) [a] | 2.0 (4.0) [b] | 0.089 | 7.5 (9.7) [b] | 1.0 (0.0) | 1.5 (4.5) [b] | 0.104 |
| Moderate | 13.0 (6.0) [a,b] | 1.5 (0.0) [a] | 2.0 (4.0) [b] | 0.089 | 7.5 (9.7) [b] | 1.0 (0.0) | 1.5 (4.5) [b] | 0.104 |
| Light | 2.5 (0.0) | 12.0 (6.0) | 8.0 (9.5) | 0.170 | 4.0 (13.2) | 5.0 (0.0) | 9.5 (15.0) | 0.138 |
| Sedentary | 0.5 (0.0) [a,b] | 34.0 (26.0) [a,c] | 11.0 (13.5) [b,c] | 0.016 | 0.0 (0.7) [a,b] | 14.0 (13.0) [a] | 10.5 (15.0) [b] | 0.007 |

Notes: IQRs = interquartile ranges [Q3–Q1]; PA = physical activity; SB = sedentary behaviour; MVPA = Moderate to vigorous physical activity; and common superscripts indicate significant differences ($p < 0.05$) between the groups with the same letter after Mann-Whitney test.

Figure 1 shows the average device-measured time (min/day, Monday to Friday) in PA levels and sedentary behaviour in both trips (to and from university) by mode of commuting. The figure reveals that in both trips (to and from university), students active commuters presents significant differences ($p < 0.05$) with both motorized modes in all PA levels and sedentary behaviour.

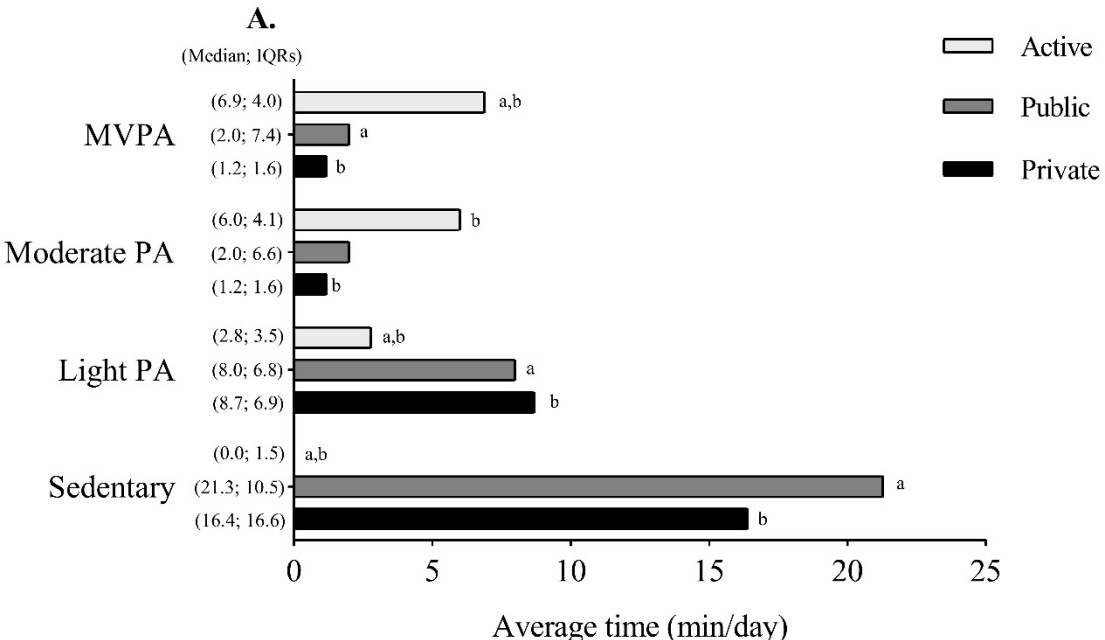

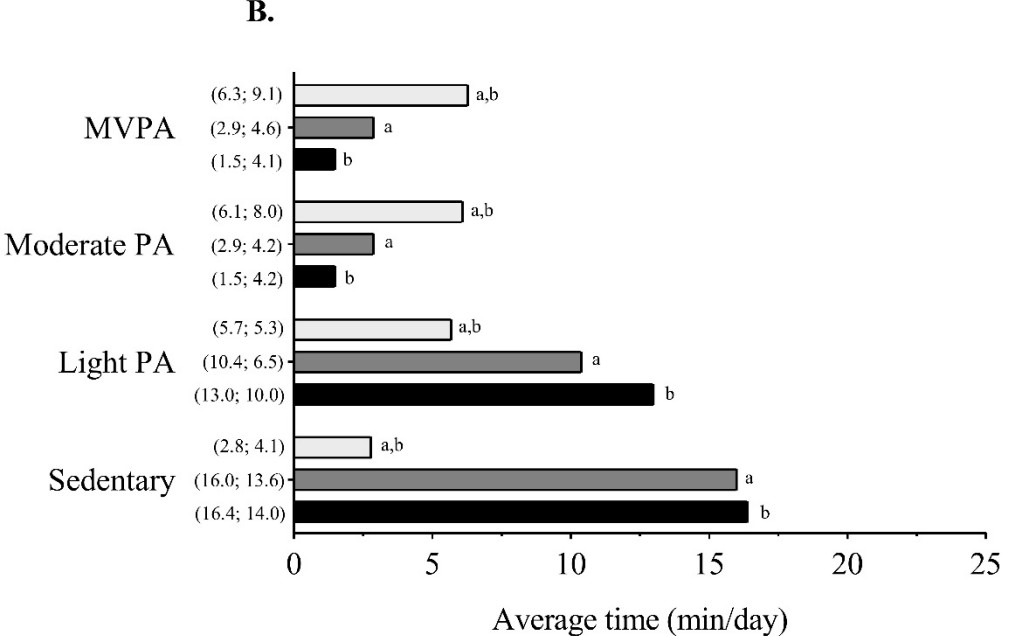

**Figure 1.** Average device-measured time (min/day, Monday to Friday) in PA levels and sedentary behaviour (**A**) to university and (**B**) from university by mode of commuting of the participants. Notes: data were expressed as medians; IQRs = interquartile ranges [Q3–Q1]; PA = physical activity; MVPA = moderate to vigorous physical activity; superscripts indicate significant differences ($p < 0.05$) between the groups with the same letter after Mann–Whitney test.

Associations between self-reported and device-measured of commuting behaviours, PA, and sedentary behaviour are available in Supplementary Material Table S2 (to univer-

sity) and Supplementary Material Table S3 (from university). On one hand (to university), students using active commuting presented positive associations in commuting time, distance and total energy expenditure with MVPA, moderate PA, and sedentary behaviour (all $p < 0.05$). Students using public commuting presented two positive associations in commuting time and total energy expenditure with light PA (both, $p < 0.05$). Private commuters reveal positive associations in commuting time, distance, and total energy expenditure with light PA, and sedentary behaviour (all, $p < 0.05$). On the other hand (from university), students using active commuting presented positive associations in commuting time with MVPA, and moderate PA, and in distance and total energy expenditure with MVPA, moderate PA, light PA, and sedentary behaviour (all, $p < 0.05$). Students using public commuting presented positive associations in commuting time with light PA and sedentary behaviour (both, $p < 0.05$). Finally, students using private commuting showed positive associations in commuting time, distance, and total energy expenditure with light PA, and sedentary behaviour (all, $p < 0.05$).

## 4. Discussion

In the current study, active commuters had higher energy expenditure per min and MVPA than public and private commuters. In this sense, in the United States, a study showed that increasing daily energy expenditure in university populations could lead to gradual and sustained improvement in cardiometabolic health [29], producing a positive balance in annual energy expenditure despite travelling shorter distances in less time [30]. The higher MVPA levels in active commuting could be an important contributor to PA recommendations [31], which would be reflected in physical health benefits [32] and fewer detrimental changes in the brain (associated with dementia later in life) [33]. The World Health Organization (WHO) recommends that adults should engage in an average of 150–300 min of moderate PA, 75–150 min of vigorous PA, or an equivalent combination of MVPA each week [32]. In this study, active commuting to and from university contributed to 66 min of MVPA weekly (compared to 24 and 13 min for the public and private commuters, respectively), which corresponds to 44% of the current MVPA recommendations. This is in accordance with results in adolescents, which showed that active commuting contributed to 36% of daily MVPA recommendations [16]. However, previous research showed that students who report having performed 40 min of MVPA per week usually consider only activities such as weight lifting, aerobic exercise, and sports (among others), but not walking [34]. In this line, active commuting could be undervalued and has been usually considered a light PA behaviour, but it should be considered as a potential MVPA with relevant contributions to physical health status and to other associated benefits. Additionally, public commuters presented higher MVPA compared to private commuters. This might be because public commuting involved longer walking times to and from stations and stops than private commuting, which was also found in a systematic review [24]. However, public and private commuters presented similar sedentary behaviour times (186 and 168 min per week, respectively), compared to the 14 min per week of active commuters. University students are inherently considered a population at risk of sedentary behaviour since a significant proportion of their time is dedicated to study or attending lessons [35]. Furthermore, high volumes of sedentary behaviour in this population have been associated with higher health risks [36]. Therefore, according to the findings, choosing between motorized options, it would be advisable to use public commuting over private commuting due to it is contribution to MVPA levels.

In this respect, it seems possible that the choice of active commuting to university, and even public commuting, could help with an issue of global importance: increasing PA levels. Nevertheless, using active and public commuting may not only be related to PA. Universities should become leaders in educating for sustainable development goals (SDGs), which are a call to action by all countries to promote prosperity and protect the planet [37]. Promoting active commuting, even public commuting, can give rise to contributing to two goals of the SDGs: Goal 3 (ensure healthy lives and promote well-being for all at all

ages) and Goal 11 (make cities inclusive, safe, resilient, and sustainable) [38] compared to private commuting. For instance, a modelling study in New Zealand estimated that active commuters could have less exposure to air pollution based on less distance travelled by private travellers [39]. According to this and the results of this study, if we estimate that there would be a reduction of 18 km driven associated with the use of private commuting, if active and public commuting were to be prioritised, this would be beneficial for the environment. Therefore, one practical implication of this is the possibility of encouraging students to use active commuting as a daily necessity to get to and from university, not only for purposes related to individual health but also seen from environmental point of view.

Following the methodological issue regarding commuting behaviour variables based on self-reported questions [7] is a possible alternative in the absence of device measures, and it has been studied as a powerful tool for managing transport in universities as well as possible effective interventions [40]. With only two self-reported questions, it is possible to estimate a quantitative measure that allows researchers to use a more sensitive variable to improve the statistical power and have more accurate results. Examining whether potential self-reported measures might be highly associated with device-measured PA is of interest for improving validity and quality assessment in future survey studies in this population. Three commuting behaviour variables (time, distance, total energy expenditure) presented the strongest associations with the device measured MVPA and sedentary behaviour in active commuters and with light PA and sedentary behaviour in public and private commuters. Active commuting is the mode that provides the highest energy expenditure per min compared to public and private commuting, and the concepts of time and distance have importance because they may strongly determine choosing or not choosing active behaviours. These findings reinforce the previous methodological issue that if commuting behaviours have been highly associated with device-measured PA, if the option is use only self-report measures is available, it could be a useful method of measurement in university populations.

Moreover, focusing on the city studied, Cádiz has urban planning issues compared to other cities in Spain. One of the main problems is the poor connectivity of the public transport system—in particular, public buses—which is reflected in the abundance and lack of control of private commuting as well as in a lack of safety for active commuting modes [41]. In addition, in several areas of the city, there are some restrictions on the use of active modes, such as cycling, which limits their choice for transport. If there are no viable options for using public commuting and the conditions for active commuting are disadvantaged, this could discourage university students. These findings may help us to understand that this Spanish city needs urban reform plans for all modes of commuting, with equal or priority conditions for active and public commuting and their benefits. However, these reflections cannot be extrapolated to the rest of the Spanish cities without first considering the characteristics of each city.

Finally, to improve future directions related to self-reported and device-measured in commuting behaviours, PA, and sedentary behaviour in both trips (to and from university) by mode of commuting, future cohort studies are necessary to confirm the findings of this study. In addition, a greater focus on air pollution exposure during different modes of commuting could produce interesting information on the health status of the university population as well as the environment.

*Strengths and Limitations*

One remarkable strength is the use of self-reported and device measures to obtain the information of the different commuting behaviours, which increases the accuracy of results in our study and provides methodological issues. Additionally, to the best of the authors' knowledge, this is the first study which used, described, compared, and associated the self-reported and device-measured of commuting behaviours, PA, and sedentary behaviour in both trips (to and from) by mode of commuting in university students. Finally, this study is subject to certain limitations: (i) the small sample size—our findings must not

be extrapolated to the entire university population in the University of Cádiz; (ii) the commuting distance could be overestimated or underestimated, as the shortest walking distance in the network was used; (iii) In addition, with respect to the recruitment of participants as well as the procedures for their participation, it is important to consider possible selection bias in the present study.

### 5. Conclusions

Walking approximately 7 min per trip (to or from university) could contribute to 44% weekly of the MVPA recommendations to obtain physical health benefits in university populations, as it involves the highest levels of energy expenditure per min, followed by public commuting. This study presents new evidence in university students from Spain with self-reported and device measures in relation to variables of commuting behaviours, PA, and sedentary behaviour. Further studies with interventions are required to examine deeper and promote this source of PA among university students, which could be a real opportunity to reduce the total volume of sedentary behaviour in university students.

**Supplementary Materials:** The following supporting information can be downloaded at: https://www.mdpi.com/article/10.3390/su142214818/s1. Table S1: Calculations of both commuting energy expenditures used based on the code of the Compendium of Physical Activities for Adults, according to the mode of commuting to university; Table S2: Associations between self-reported variables of commuting behaviours to university and device-measured time of PA, and sedentary behaviour.; Table S3: Associations between self-reported variables of commuting behaviours from university and device-measured time of PA, and sedentary behaviour.

**Author Contributions:** Conceptualization, X.P.-L. and P.C.; data curation, X.P.-L., V.S.-J., A.P.-B., A.S.-D., and D.C.-M.; formal analysis, X.P.-L. and D.C.-M.; funding acquisition, X.P.-L., D.C.-M. and V.S.-J.; investigation, A.P.-B. and A.S.-D.; methodology, X.P.-L., P.C. and V.S.-J.; project administration, D.C.-M.; supervision, P.C., V.S.-J. and D.C.-M.; writing—original draft, X.P.-L.; writing—review and editing, P.C., V.S.-J., A.P.-B., A.S.-D. and D.C.-M. All authors have read and agreed to the published version of the manuscript.

**Funding:** This research was funded by the National Agency for Research and Development (ANID)/Scholarship Program/DOCTORADO BECAS CHILE/2020–(Grant N° 72210020) for X.P.-L. and was supported by the University of Cádiz, Plan Propio de investigación 2017 (PR2017-087) for D.C.-M. Additionally, V.S.-J. was funded by Instituto de Salud Carlos III through the fellowship CP20/00178 co-funded by European Social Fund.

**Institutional Review Board Statement:** The study was conducted in accordance with the Declaration of Helsinki [42] and was reviewed and approved by the Ethics Committee at the University of Cádiz, Spain (Ref. 004/2021).

**Informed Consent Statement:** Informed consent was obtained from all subjects involved in the study.

**Data Availability Statement:** All data are available digitally and are available on request.

**Acknowledgments:** The authors would like to thank all the students and the institutions who participated and collaborated in this study for making this research possible.

**Conflicts of Interest:** The authors declare no conflict of interest, financial or personal, that would influence the work presented in this study.

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
