# Peer review of "Commuting to University: Self-Reported and Device-Measured Physical Activity and Sedentary Behaviour"

_sustainability, doi:10.3390/su142214818_

Round 1

Reviewer 1 Report

Title

Overall, the title of the manuscript reads well and might spot interest in the reader. My only concern is that the title might be too long. Authors should consider shortening the title, for example I am not sure if it is necessary to say, “Commuting to university” or “The UCActive study”.

Abstract

Overall, the abstract is well written. My only concern is that the results are very generalized. The presentation of results could be more specific.

Introduction

The introduction is accurate and provides the most updated information about issues of physical activity levels. However, the introduction is clearly too short, and Authors should study more recent research on this topic. One of the main aims of this study is to compare the self-reported and device-measured physical activity. There are several recent research papers that have pointed out, that there is a great correspondence issue between levels of self-reported and device-based physical activity (for example, see a study by Kalajas-Tilga et al., 2022). The inclusion of such studies could strengthen the introduction of this study.

Kalajas-Tilga, H., Hein, V., Koka, A., Tilga, H., Raudsepp, L., & Hagger, M. S. (2022). Trans-Contextual Model Predicting Change in Out-of-School Physical Activity: A One-Year Longitudinal Study. European Physical Education Review, 28(2), 463–481. https://doi.org/10.1177/1356336X211053807

Materials and Methods

Authors could explain why the sample size was so small.

Results

Overall, the results are well presented. My only concern is the readability of the figures. Please improve the quality of figures.

Discussion

The discussion is an accurate interpretation of the study results. However, the discussion is clearly too short. Authors are recommended to add more discussion by comparing the results with other previous studies.

Please provide more practical implications of the current study.

Please provide more strengths and limitations of the current study.

Please provide more future directions based on the current study.

Author Response

Reviewer 1

We would like to thank the reviewers for their thoughtful and constructive comments and feedback. We have considered all the suggestions and have incorporated them into the revised manuscript.

A new revision of our manuscript, "Commuting to university: self-reported and device-measured physical activity and sedentary behaviour", is attached, along with responses to your suggestions below. We believe our manuscript is stronger because of these revisions.

Cordially,

The authors.

Reviewer 2 Report

Dear Authors, congratulations on your work.

Despite being well structured, in my point of view, it needs some clarification:

Introduction

Turn the objectives clear. Is confusing to read the goals of the work and read the data present in the results section. It is confusing to readers. I noticed that in table 3 appears info that isn't expected, because where is the first time that you write about commuting per weekday. This analyse isn't the accord with the goals of this work. In my point of view, of course. Please clarify.

The authors don't make reference to various types of commuting in the introduction section. Should appear a paragraph that abords the difference of the diverse type of commuting.

Methods

Clarify better the use/location of the accelerometer. Confusing information about this...Line 140: The appliance was placed around the hip and worn for 8 days; Line 149: They were instructed to wear the accelerometer on their lower back, which was attached by an elastic belt.

Clarify whether the Questionnaire of mode of commuting and PA to the university is validated for the Spanish reality.

Describe the assumptions for the use of post-hoc Bonferroni.

Discussion

Bearing in mind that this study was developed in a specific location, I think it is important to reflect on the impact of the location on the results, and if these can be transversal to the rest of the student Spanish population.

 Regards,

Author Response

Reviewer 2

We would like to thank the reviewers for their thoughtful and constructive comments and feedback. We have considered all the suggestions and have incorporated them into the revised manuscript.

A new revision of our manuscript, "Commuting to university: self-reported and device-measured physical activity and sedentary behaviour", is attached, along with responses to your suggestions below. We believe our manuscript is stronger because of these revisions.

Cordially,

The authors.

Round 2

Reviewer 1 Report

Authors have done well job on revising the manuscript.

Author Response

Response to reviewer comments

Thank you for your comments as an expert reviewer to improve the manuscript, all the suggestions have been helpful to us as we continue to learn as researchers. 

We believe our manuscript is more substantial because of these revisions.

Cordially,

The authors.

Reviewer 2 Report

Dear author,

thank you for addressing my comments.

Congratulations.

Author Response

(The authors gave the same response as above.)
